 

🔓 | **Open Peer Review** | Host-Microbial Interactions | Research Article

# A sequential one-pot approach for rapid and convenient characterization of putative restriction-modification systems

Yi Zhang,[1] Yoshihiro Takaki,[1] Yukari Yoshida-Takashima,[1] Satoshi Hiraoka,[2] Kanako Kurosawa,[1] Takuro Nunoura,[2] Ken Takai[1]

**ABSTRACT** With the advance of high-throughput sequencing, the molecular basis of coevolutionary interaction between viruses and host microorganisms is predominantly elucidated by *in silico* genomic analyses, which revealed potential communication of genetic materials related to microbial immune systems such as the restriction-modification (R-M) system. However, the sequence-dependent information is often insufficient to output a conclusive argument without biochemical characterizations, particularly for homologs of rare genes considered less accurately annotated. We proposed a 1-day and one-pot workflow covering *in vitro* protein synthesis and enzymatic assays to confirm the exact function of putative R-M genes only with manual pipetting operations of microliter-scale liquids. The proof-of-demonstration experiments mainly focused on a series of putative R-M enzymes from our recently found deep-sea temperate bacteriophage and its host bacterium. Two new restriction endonucleases and two new methyltransferases with respective unambiguous substrate specificities, superior catalytic performance, or unique sequence preferences were quickly identified. A frequent discrepancy between sequence similarity search and single-molecule methylation-sensitive sequencing toward the prediction of recognition motifs can get settled with the established direct biochemical characterization. The proposed approach under the cell-free one-pot concept allows for preliminary characterizations of diverse categories (e.g., Types I, II, and III) of putative R-M systems at most laboratories with minimum equipment and time costs.

**IMPORTANCE** The elucidation of the molecular basis of virus-host coevolutionary interactions is boosted with state-of-the-art sequencing technologies. However, the sequence-only information is often insufficient to output a conclusive argument without biochemical characterizations. We proposed a 1-day and one-pot approach to confirm the exact function of putative restriction-modification (R-M) genes that presumably mediate microbial coevolution. The experiments mainly focused on a series of putative R-M enzymes from a deep-sea virus and its host bacterium. The results quickly unveiled unambiguous substrate specificities, superior catalytic performance, and unique sequence preferences for two new restriction enzymes (capable of cleaving DNA) and two new methyltransferases (capable of modifying DNA with methyl groups). The reality of the functional R-M system reinforced a model of mutually beneficial interactions with the virus in the deep-sea microbial ecosystem. The cell culture-independent approach also holds great potential for exploring novel and biotechnologically significant R-M enzymes from microbial dark matter.

**KEYWORDS** protein function, virus-host interactions, protein-DNA interactions, restriction-modification system, fluorescence assays, enzymes, DNA methylation

Microorganisms in a given habitat form a complex interaction network in different hierarchies, most represented by the binary interaction with viruses (1, 2).

Address correspondence to Yi Zhang, zhangyi@jamstec.go.jp.

The authors declare no conflict of interest.

See the funding table on p. 15.

High-throughput sequencing platforms and the associated computational methods fully pipelined for the prokaryote studies advanced our understanding of the complex coevolutionary interaction network involving viruses (3, 4). They not only reveal the species composition but also deduce the probable way by which the members of the microbial communities interact with each other (5). As one of the underlying molecular mechanisms, the restriction-modification (R-M) system serves as a ubiquitous prokaryotic immune system and plays a major role in mediating the coevolutionary interaction through selectively cleaving or methylating the bilateral genetic materials (6–9). The species diversity involved in the virus-host interaction in nature implies the still-growing molecular diversity of restriction endonuclease (REase) and methyltransferase (MTase), while the diverse enzyme activities and substrate preferences at least partially underlie the dynamic epigenomes of prokaryotes (10). In contrast to CRISPR-associated (Cas) nucleases, whose recognition sequence specificity is straightforwardly determined on demand by individual guide RNA molecules (11), the recognition sequence specificity of the R-M system depends on respective R-M enzymes, whose elegant sequence-structure-function relationship has not yet been fully elucidated (12). Therefore, careful individual characterizations of enzyme activity and sequence specificity are generally required for newly predicted REases/MTases of interest (13). However, the genomic sequence-dependent information is often ambiguous and generally less conclusive because of the lack of exact functional confirmation (4, 14, 15).

Direct biochemical characterization is the sole effective means to clarify such ambiguities. However, the heterologous protein expression, which has undoubtedly scored many great successes with elaborate and elegant system optimization (16), is generally not easy for the R-M enzymes because of their intrinsic cytotoxicity in host organisms. The lengthy and labor-intensive protein purification is also a major obstacle to the rapid characterization of protein functions. It can be significant to the sequence-function relationship studies if a series of experiments from protein expression to function confirmation could be done within a reasonably short time and potentially compatible with batch processing. With the advance in synthetic biology tools, we think of a convenient way to bypass protein purification with a fully reconstituted cell-free protein synthesis (CFPS) system for the preparation of R-M enzymes, which is termed PURE (protein synthesis using recombinant elements) and is only composed of the minimum set of molecules required for transcription and translation reactions (17). The PURE system intrinsically fulfills the demand of being (i) free of cytotoxicity and (ii) plausibly free of components potentially interfering with subsequent protein assays. The first characteristic is common among various cell-free expression systems (18–20), while the second characteristic is unique to the PURE system and worthy of maximizing this advantage beyond CFPS.

In the present study, we pursued simplicity to enable the experimental operations of protein synthesis and function characterization to be only composed of pipetting liquids in a single reactor, akin to the "one-pot" strategy in chemical synthesis by which sequential reactions proceed without product purification at every step. With the new technique, we successfully characterized a putative R-M system encoded in our recently found temperate bacteriophage (or phage for short) infecting a deep-sea *Campylobacterota*, *Nitratiruptor* bacterial strain. Previously, the genomic and epigenomic analyses figured out complex coevolution between the deep-sea *Nitratiruptor* hosts and their temperate phages (21), while complete R-M systems have so far been considered nearly absent in phages (22). However, the sequence recognition of the R-M system in the virus-host interaction estimated by a sequence similarity analysis could not explain the epigenetic signatures despite abundant research activities around the biotechnologically significant Type II R-M system (23–25). The one-pot approach is a promising tool to prove complex coevolution among microbes, viruses, and other mobile genetic elements, as well as to explore novel biotechnological tools in the era of epigenomics.

## RESULTS

### One-pot synthesis and characterization of putative restriction endonuclease

Our previous work predicted the presence of a complete Type II R-M system [NRS3_07 as the MTase, GenBank: BCD83123.1 (26); NRS3_08 as the REase, GenBank: BCD83124.1 (27)] in a temperate phage NrS-3 infecting a *Campylobacterota* strain *Nitratiruptor* sp. YY08-14 (NCBI: txid2724899) isolated from a deep-sea hydrothermal vent ecosystem (Fig. S1) (21). The previous study annotated the putative REase NRS3_08 as BglII based on sequence similarity, whereas the parallel SMRT (single-molecule real-time) sequencing detected almost 100% RG($^{6m}$A)TCY methylation ($^{6m}$A or 6mA: N6-methyldeoxyadenosine) across both the phage and bacterial genomes, which does not match the sequence specificity (i.e., AGATCT) and the DNA methylation sensitivity of the homologous BglII. Thus, the recognition sequence of the putative R-M system still remains undetermined. To determine the sequence specificity, a series of fluorogenic molecular beacon (MB) probes containing each of all the probable recognition sequences, AGATCT (MB-1 probe), GGATCC (MB-2 probe), AGATCC (MB-3 probe), and GGATCT (MB-4 probe) (Fig. S2A), was designed and used for the fluorogenic enzyme assay (28).

First, the putative REase NRS3_08 was synthesized with the PURE system (17) in a microtube and directly used as the enzyme source without sophisticated protein purification, generating our initial concept of the so-called one-pot characterization approach (Fig. 1A). Each MB probe was mixed with the NRS3_08 CFPS solution and reaction buffer in the microwell of a 384-well microplate, and a time-course measurement was carried out. Given no background fluorescence from the negative control, the evident increase of fluorescence intensity with time for all four MB probes can only be attributed to the restriction activity of the NRS3_08 CFPS product (Fig. 1B). NRS3_08, which contains a conserved Endonuc-BglII domain (pfam09195) detected by both Conserved Domain Search (29) and InterProScan (30), was thus proven to be an REase targeting the sequence RGATCY. The homologous BglII only cleaved the MB-1 probe (Fig. 1C) and thus showed a cleavage specificity distinct from NRS3_08. The one-pot experimental characterization supported the prediction based on the SMRT sequencing data. As the biochemical function was exactly confirmed, according to a nomenclature (31), the NRS3_08-encoded REase (provisionally registered as Nsp0814ORF332P in REBASE) was named R.Nph3I.

For comparison purposes, two commercially available isoschizomers with opposite DNA methylation sensitivities, MflI and BstYI, were tested with the same set of MB probes in parallel. MflI sensitive to the 6mA DNA methylation showed a set of cleavage kinetics clearly different from R.Nph3I (Fig. 1D). Intriguingly, BstYI, which is not sensitive to the 6mA DNA methylation, showed a cleavage kinetics profile identical to R.Nph3I (Fig. 1E). Due to the asymmetric secondary structure of the DNA substrate and potential inaccuracy of the concentration determination for fluorescently modified and hairpin-structured DNA oligos, the different cleavage kinetics (e.g., the initial rate of the cleavage reaction) for a given REase toward different MB probes may not be simply equated to its recognition sequence preference (see Discussion later). Regardless, this does not compromise the primary purpose of the cleavage assay, the confirmation of the REase activity.

Since the MB probe is supposed not to interfere with the CFPS reaction, we further considered coupling the CFPS reaction with the fluorogenic restriction reaction to rapidly confirm the enzymatic activity. Individual MB probes were pre-mixed with the CFPS reaction solution. After a short delay of about 30 min, which allowed enough mature REase molecules to be synthesized in microwells, a sigmoidal increase in fluorescence intensity was detected (Fig. 1F). The intensity profile of each probe well reflected the aforementioned cleavage kinetics of R.Nph3I.

Given a potential degradation of the template DNA containing an RGATCY site within the coding sequence (CDS) by the expressed REase, plasmid DNA was used as the template for the sustained protein expression of R.Nph3I. To further reduce the overall

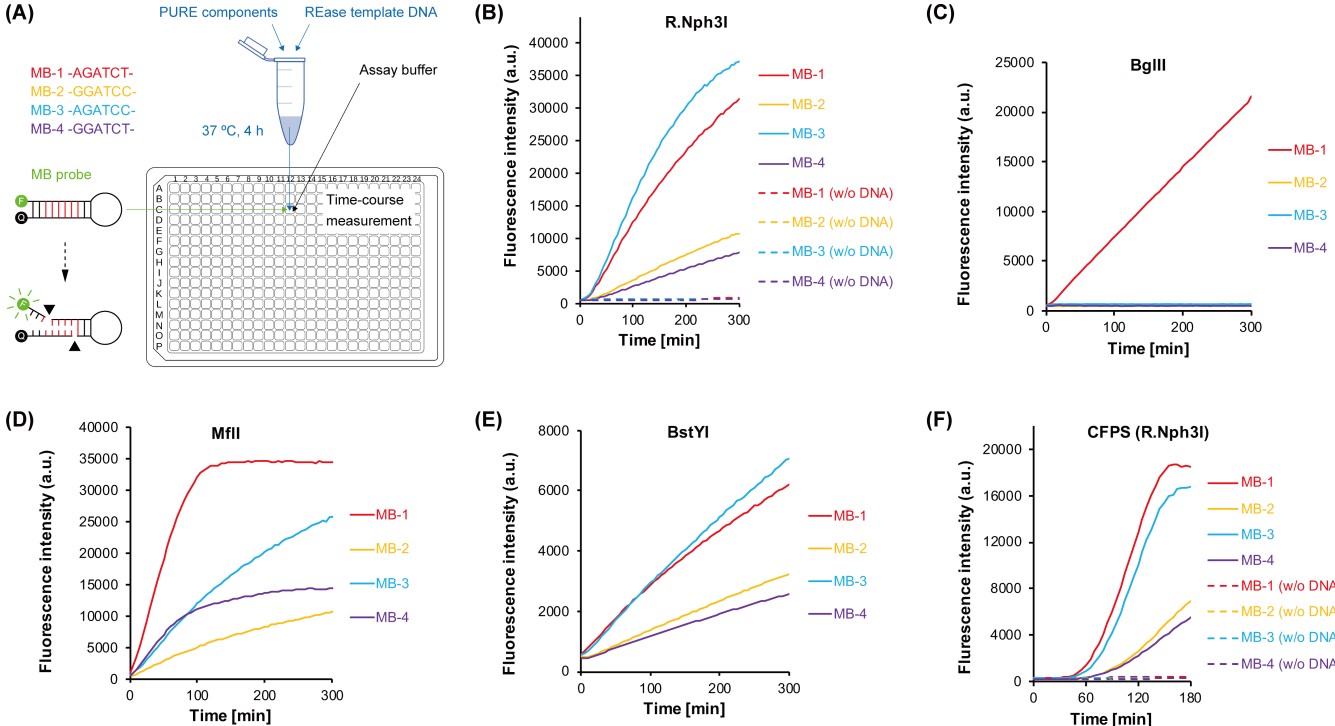

**FIG 1** A general one-pot approach for restriction endonuclease (REase) characterizations. (A) The target REase synthesized in a microtube by the PURE system is mixed with the assay reagents containing a specific molecular beacon probe. The MB probe is a short hairpin-shaped DNA oligo with a terminal fluorophore FAM (F) at one end and a terminal quencher BHQ-1 (Q) at the other end. The fluorescence of FAM is repressed by the nearby BHQ-1 and can be restored when the MB probe is cleaved by functional REase. (B) Cleavage activity and specificity assays for the putative REase R.Nph3I from phage NrS-3 infecting the *Nitratiruptor* sp. YY08-14 strain. Individual MB probes (MB-1, 2, 3, and 4 representing each sequence of the predicted motif RGATCY; see Fig. S2A) were mixed with the final CFPS solution. As negative controls, a mixture of the PURE components without the template DNA was also reacted with individual probes. (C-E) Cleavage specificity assays for BglII, MflI, and BstYI, respectively. The same MB probes were used as the enzyme substrate. (F) CFPS coupled with the MB fluorogenic reaction. The negative control groups without the addition of the template DNA were labeled as "w/o DNA". a.u., arbitrary unit.

experiment time when the use of plasmid can be considered unnecessary, we proposed using overlap extension PCR a ligation-independent technique to fuse multiple DNA fragments seamlessly (32), to prepare linear template DNA compatible with the PURE system (Fig. S3). As a proof of demonstration, we arbitrarily characterized another putative REase mel_015 (GenBank: AIT54628.1 (33), REBASE: Mis1ORF16P) presumably targeting GATC, which was recently predicted from the genome of *Marseillevirus* isolated from *Acanthamoeba castellanii* (34). The resulting CFPS solution was directly used for the restriction assay with an arbitrary one of the aforementioned MB probes that contained the GATC motif (Fig. S4). The single-day experiment covering the template preparation, protein synthesis, and enzymatic assay proved that mel_015 was an REase targeting GATC. The mel_015-encoded REase was named R.Mis1I, following a convention of an already-named cognate MTase (M.Mis1I).

## DNA methylation sensitivity determination

Since the previous SMRT sequencing revealed fully (i.e., 100%) methylated RG($^{6m}$A)TCY motif in the genomes of *Nitratiruptor* sp. YY08-14 and its phage NrS-3 (21), the REase R.Nph3I likely constitutes an active R-M system with the adjacent putative MTase NRS3_07 and is predicted to be sensitive to DNA methylation. To verify this hypothesis, we applied site-specifically methylated MB probes to determine the methylation sensitivity of R.Nph3I with the one-pot approach (Fig. 2A).

The MB probe was chemically synthesized using usual unmethylated A, T, G, and C, except for the two A of the double-strand RGATCY recognition site, which were

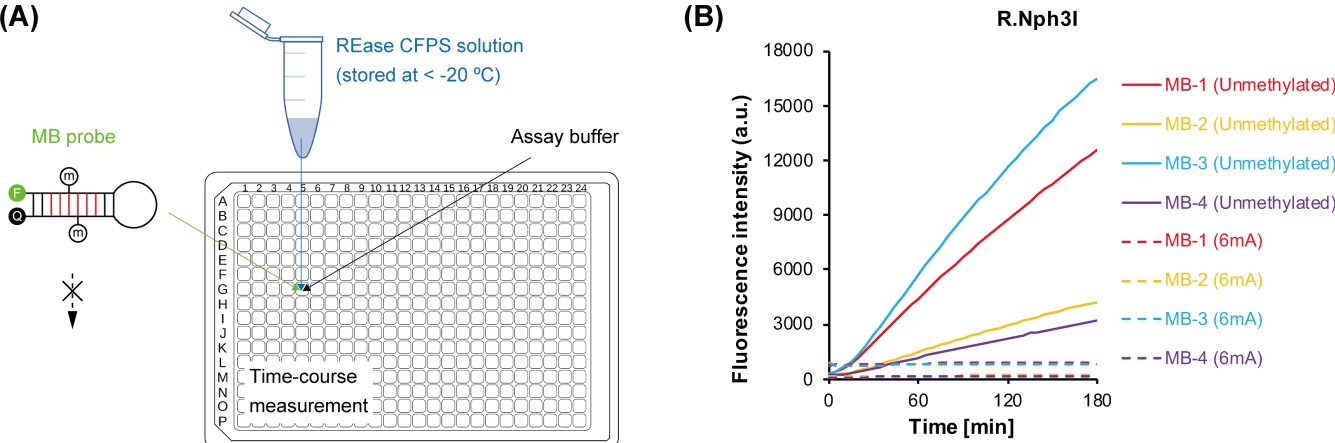

**FIG 2** One-pot experimental characterization of DNA methylation sensitivity. (A) Restriction cleavage of MB probes blocked by DNA methylation. The methylated MB probe is mixed with the REase solution to confirm if the probe can be cleaved. (B) Cleavage activity of R.Nph3I toward unmethylated or fully methylated DNA. The methylated DNA probes were chemically synthesized using N6-methyladenine (6mA). Individual MB probes in equal quantity were subjected to the restriction reaction.

synthesized using N6-methyladenine (6mA). A complete inhibition of the REase activity was observed (Fig. 2B), proving that R.Nph3I is sensitive to the 6mA DNA methylation. Similarly, another restriction assay using the same 6mA-modified MB probe proved that R.Mis1I is also sensitive to the 6mA DNA methylation (Fig. S4).

Overall, the proposed approach refined the bioinformatic predictions about the recognition sequence and the methylation sensitivity and provided additional information about the cleavage kinetics of R.Nph3I.

## One-pot synthesis and recognition sequence-dependent characterization of putative methyltransferase

To confirm the MTase activity, NRS3_07, the other half of the putative R-M system, was also synthesized using the PURE system, and the resulting CFPS solution was directly used as the enzyme source without protein purification. We tried S-adenosylmethionine (SAM) as the methyl donor and applied a continuous enzyme-coupled assay to detect the MTase activity (35). The hypothetical methylation reaction for the MB probes by NRS3_07 was coupled with a methyl-directed restriction reaction by DpnI (only cleaving G($^{6m}$A)TC), resulting in fluorescence (Fig. 3A). The result supported that NRS3_07 indeed served as an MTase capable of methylating RGATCY to RG($^{6m}$A)TCY (Fig. 3B). The NRS3_07-encoded MTase (provisionally registered as M.Nsp0814ORF332P in REBASE) was named M.Nph3I.

The expressed MTase does not methylate its CDS because of the absence of RGATCY sites on it. Therefore, the methyl-directed restriction reaction by DpnI does not target the template DNA and is supposed to be orthogonal to the CFPS reaction. Similar to the demonstration in Fig. 1F, we next tried coupling the CFPS reaction with the probe methylation reaction and the DpnI cleavage reaction in a single microwell. The MTase activity was readily detected after a short delay, while a negative control without DpnI showed no background (Fig. S5). The all-in-one-pot approach greatly simplified the initial preliminary confirmation of the REase/MTase activities, provided that the orthogonal reaction settings can be satisfied.

## Interplay between M.Nph3I and R.Nph3I

Since the enzyme activity and sequence specificity were individually clarified for both R.Nph3I and M.Nph3I, an *in vitro* rapid prototyping of the integrated R-M system becomes feasible. The experimental workflow was similar to the methylation sensitiv-

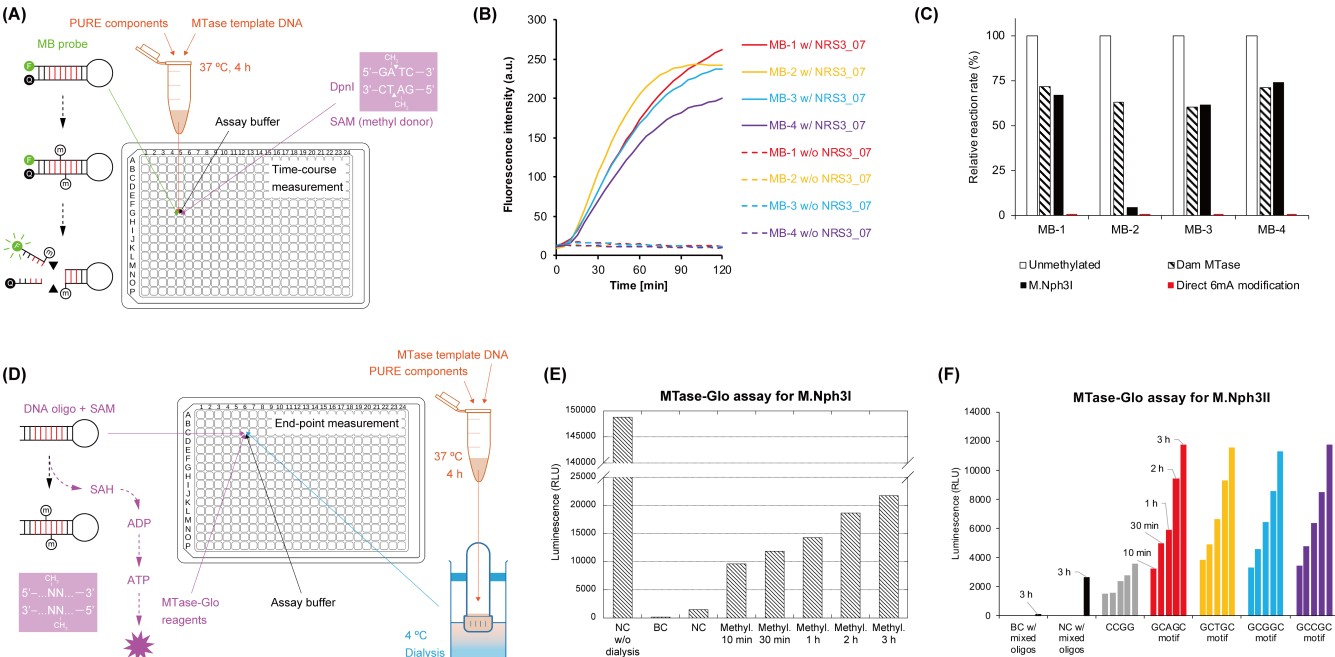

**FIG 3** One-pot characterization of putative methyltransferase (MTase) activity. (A) Methyl-directed restriction cleavage depending upon DNA methylation of the MB probe. In this case, DpnI can only recognize and cleave the G($^{6m}$A)TC motif pre-generated by the putative MTase. (B) Methylation activity and specificity assays for the putative MTase M.Nph3I (i.e., NRS3_07). The MB probes without the DNA methylation treatment were also subjected to the DpnI restriction reaction (dashed lines). (C) Relative initial reaction rates of R.Nph3I for each MB probe prepared by different methods. The reaction rate toward unmethylated probes was normalized to 100% (hollow columns). In contrast to the unmethylated (filled with none) or 6mA-chemically modified (filled with red color) groups, in which the MB probes are pure single components with well-defined methylation states, the probe solution enzymatically prepared by either Dam MTase (filled with diagonal stripes) or M.Nph3I (filled with black color) contains both methylated and unmethylated probes as a mixture. The initial rate of the restriction reaction is dependent on the concentration of unmethylated MB probe in the mixture. (D) Sequence-independent MTase assay using dialyzed CFPS sample. The dialysis removes ATP, ADP, and AMP in the CFPS solution, and the assay converts the by-product (SAH) of the DNA methylation reaction to ATP and detects ATP using luciferase and light-emitting compound. (E) MTase-Glo assay results for samples without and with dialysis. BC, blank control (i.e., the assay buffer only); NC, negative control (i.e., the PURE components mix without the template DNA); Methyl. x min or h, DNA methylation reaction for different lengths of time. (F) MTase-Glo assay for the putative MTase M.Nph3II. Four DNA substrates constituting the candidate GCNGC motif (Fig. S2C) reacted with the dialyzed MTase sample and SAM for 10 min, 30 min, 1 h, 2 h, and 3 h. An arbitrary DNA substrate with CCGG recognition site was also subjected to the assay with the dialyzed MTase (gray colored). A negative control sample (dialyzed PURE mix without template DNA) and a blank control sample (buffer only) were also prepared and reacted with the mixture of those four DNA oligos and SAM for 3 h.

ity assay but applied MB probes methylated by the MTase instead of the chemically synthesized ones. In brief, the MB probes were methylated using the M.Nph3I CFPS solution and then subjected to the restriction reaction using the R.Nph3I CFPS solution.

The initial rate of restriction reaction is correlated with the concentration of unmethylated DNA, which can be used to estimate the methylation efficiency of MTase. Surprisingly, the interplay between M.Nph3I and R.Nph3I revealed a strong sequence preference of M.Nph3I toward GGATCC (MB-2) (Fig. 3C). Over 95% of the GGATCC motif was methylated by M.Nph3I, while the other three motifs were only 32 ± 6% methylated. For comparison, the same MB probes were also subjected to DNA methylation with *E. coli* Dam MTase [methylating GATC to G($^{6m}$A)TC] (Fig. S6). In sharp contrast to M.Nph3I, Dam did not show such sequence preference and modified all four sequences with a similar methylation efficiency of 33 ± 6% (Fig. 3C).

## Recognition sequence-independent characterization of putative methyltransferase

The above MTase assay relies on DpnI, while most putative MTases possess diverse sequence specificities and methylation chemistries (36, 37), which cannot be targeted by

any available methyl-directed REase (http://rebase.neb.com/rebase/azlist.md2.cy.html). To expand the scope of the one-pot approach toward any given MTase, we next tried coupling a chemiluminescent assay called MTase-Glo (38), a universal method capable of converting the by-product unique to the DNA methylation reaction, S-adenosylhomocysteine (SAH), to adenosine triphosphate (ATP) and detecting the ATP with a well-known luciferase catalysis chemistry (Fig. 3D).

We first applied M.Nph3I and its preferred substrate (namely, oligo-1, which has the same DNA sequence as the MB-2 probe but without terminal modifications; see Fig. S2B) to establish the workflow. Before performing the MTase-Glo assay, the final CFPS solution of the M.Nph3I was subjected to microdialysis, by which 99% of internal ATP and its derivatives, ADP and AMP, were removed (Fig. 3E). The dialyzed sample was then directly mixed with the substrates (DNA and SAM) and the assay reagents in the 384-well microplate. Strong chemiluminescence well above the minimal background was successfully detected for the dialyzed M.Nph3I sample (Fig. 3E). Besides, the luminescence intensity was proportional to the methylation reaction time: the longer the reaction time, the higher the concentration of accumulated SAH and the higher the luminescence intensity obtained. In this workflow, the one-pot feature was maintained as the only required operation for the additional microdialysis was no more than dispensing liquid in and out of a single dialyzer.

The SMRT sequencing in our previous study detected not only the RGATCY motif but also a GCNGC motif (21). Since no additional MTase-like coding sequences were detected in the viral genome, the DNA methylation of the GCNGC motif was predicted to be catalyzed by a putative MTase C0176 [GenBank: BCD63364.1 (39)], which was deduced from the *Nitratiruptor* sp. YY08-14 host. However, the non-putative top hits (identity >10%) in a BLASTP search against the REBASE database were all 5mC (i.e., 5-methylcytosine) MTases with a different sequence specificity of GCSGC (M.CglI, M.NgoFVII), GCCGC (M.NgoAVII), GCWGC (M.LmoJ2I, M.Gsp300IV, M.TcoKWC4III, M.Bve291I, M.BamWS8I, M.CthV, M.TacII, M.EfaRFI, and M.Saf8902II), GCAGC (M.Lsp1109I, M.BceSIV), GCNNGC (M1.BscXI, M2.BscXI), GCGC (M.Cje11351IV), CCGG (M.Ccel, M.Csp68KIV), or GGCC (M.Bfi10335I, M.HaeIII, M.FnuDI, M.Pmu384III, and M.Tvu2HI), but never GCNGC. To verify our hypothesis, the putative MTase C0176 was synthesized, dialyzed, and subjected to the MTase-Glo assay with each of four hairpin-shaped DNA oligos representing the GCNGC motif (Fig. S2C). The MTase-Glo assay revealed an MTase activity of C0176 targeting GCNGC (Fig. 3F). The C0176-encoded MTase (provisionally registered as M.Nsp0814ORF176P in REBASE) was named M.Nph3II. A methyl-directed restriction assay under the one-pot scheme further suggested that M.Nph3II should be a 5mC MTase (Fig. S7). An independent *in silico* reanalysis of our previous PacBio sequencing data was also in agreement with this experimental result (Table S1).

## DISCUSSION

### Brief overview of the one-pot workflow

Fig. 4A streamlined the workflow for the experimental characterization of putative R-M enzymes. It is composed of three primary steps: template DNA preparation, cell-free protein synthesis, and enzymatic assays. Additional steps, such as protein dialysis, can be modularly incorporated into the pipeline as required without compromising the one-pot characteristic. Compared with the conventional heterologous protein expression and purification, the one-pot design greatly simplified the experimental operation, lowered the labor intensity (or eased labor shortages in another sense), and reduced the reagent consumption. The one-pot approach we proposed herein mainly referred to liquid dispensing, while the state-of-the-art and fast-growing robotic technologies have recently allowed rather complex operations (40–43). The combination of the one-pot approach and some customizable robotic systems allows for full automation of the entire workflow, from the initial template preparation to the functional assays. In cooperation with culture-dependent epigenomic and culture-independent metaepigenomic analyses

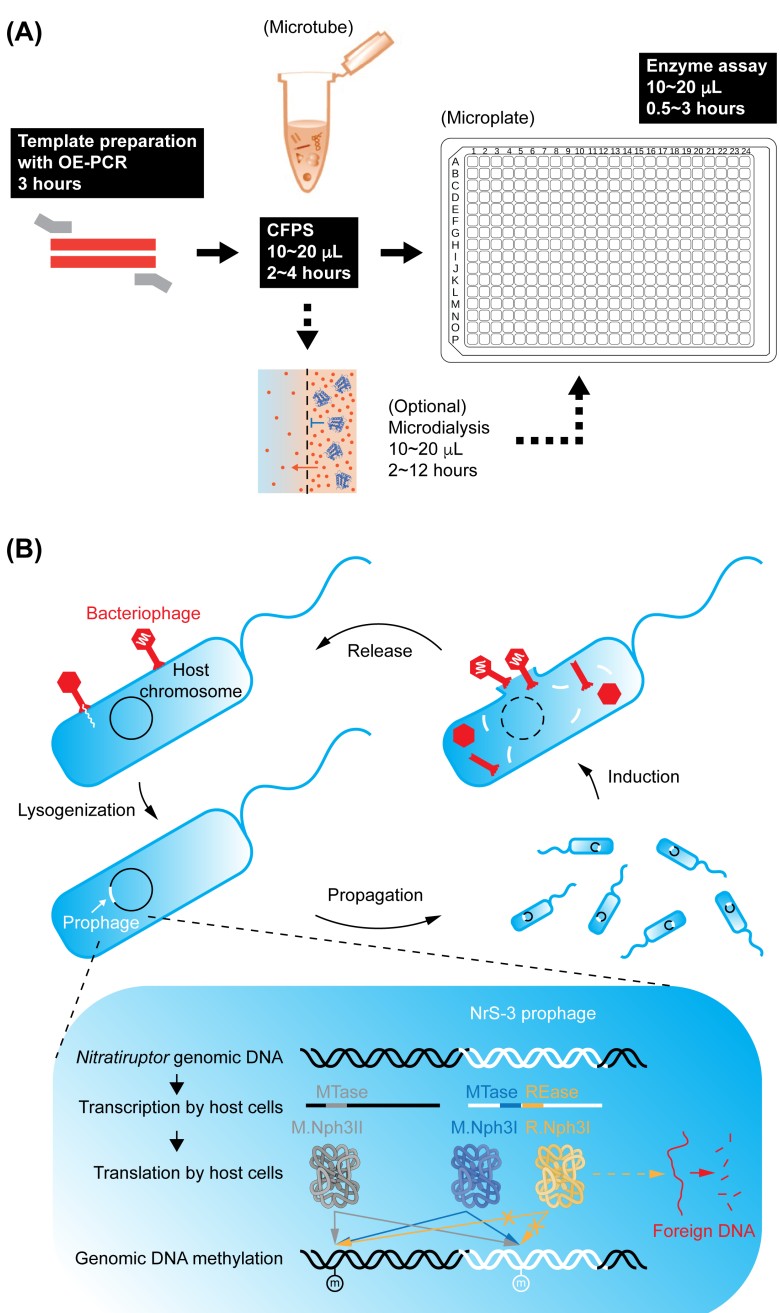

**FIG 4** (A) Summary schematic of the sequential workflow for the one-pot characterization of putative R-M enzymes. The entire process, including template DNA preparation, can be done in a day and is compatible with high-throughput and miniaturized apparatuses for protein synthesis, sample pre-treatment, and functional assays. No additional operations other than pipetting and incubation are required. (B) A tentative model about the role of the R-M systems (if not all) in the deep-sea phage (NrS-3)-host (*Nitratiruptor*) interactions. The lysogeny may confer a benefit to the host through the prophage-encoded REase R.Nph3I that might protect against invading foreign DNA. The prophage-encoded MTase M.Nph3I may protect the bacterial and phage genomes from digestion by R.Nph3I. The prophage genome may also receive protection from the host MTase M.Nph3II. The arrows and the methylation markers do not indicate a specific methylation target site but the position of all possible methylation sites, which is located on either the host side (black-colored DNA) or the prophage side (white-colored DNA), representing the phage-host interaction.

based on SMRT sequencing (16, 37, 44), the one-pot approach would be a promising method to evaluate vast R-M genes originating from diverse prokaryotes and viruses not yet cultured. Subsequently, the technique may help to unveil the coevolution among host organisms, viruses, and other mobile genetic elements encoding R-M systems.

## Biological implications from the *Nitratiruptor* case study

As the first case in the deep-sea *Campylobacterota* bacterium in combination with our previous *in vivo* characterization of its epigenetic features (21), the reality of the functional R-M system that was experimentally confirmed *in vitro* in this study reinforced the tentative model of mutually beneficial interaction between phage and its host (Fig. 4B). In brief, (i) the viral REase may protect the host by digesting invading foreign DNA; (ii) the cognate viral MTase may protect the host genome from digestion by the same viral REase; (iii) the prophage may be protected by the host-side MTase. There was only 1 GGATCC site among 23 RGATCY sites in the NrS-3 viral genome. Intriguingly, the sole GGATCC site was located in the CDS of the gene encoding REase R.Nph3I. Such high methylation selectivity of M.Nph3I toward GGATCC suggested its role of preferentially protecting its cognate gene from digestion by the expressed REase. This protection would benefit a sustained expression of the REase during the virus-host interaction. The expressed R.Nph3I also does not target the M.Nph3I gene because of the absence of RGATCY sites of the latter. This R-M system appeared to be perfectly pre-evolved as an individual mobile genetic element, thus supporting the idea of horizontal gene transfer of the R-M system (45, 46).

## Use of hairpin DNA probes in R-M enzyme activity measurement

MB-3 and MB-4 probes exhibited different cleavage kinetics for R.Nph3I (also for BstYI) (Fig. 1B and E), although the recognition sequences (AGATCC and GGATCT) are complementary to each other and would default to identical sites (47). In addition to the probable reason being the quantification inaccuracy of the special DNA oligos (Fig. S8), another reason we have considered is the intrinsic asymmetry of the hairpin structure of the MB probe. The DNA hairpin has a double-stranded "stem" and a single-stranded "loop," plausibly making binding, cleavage, or dissociation toward a non-palindromic DNA motif sensitive to its orientation. In particular, as the length (2 bp) of the flanking sequence in the stem region was short (could not be shortened further) in our design, the orientation effect might be a variant of a well-known flanking effect affecting cleavage efficiency. In other words, the sequence-identical motif was placed in two highly different contexts in MB-3 and MB-4, respectively, which would be very unlikely if the substrate was a common DNA strand such as linear dsDNA or plasmid DNA. As a piece of indirect evidence, the Gibbs free energy change ($\Delta G$) calculated by Oligo 7 was different between MB-3 ($\Delta G = -12.0$ kcal/mol) and MB-4 ($\Delta G = -11.6$ kcal/mol) and thus suggested that those two hairpins would not be fully identical in thermodynamic stability. The structural flanking effect sounds plausible but would require further experimentation. Also, as a prerequisite for quantitative measurement of enzyme kinetics, a convenient and reliable quantification method for chemically modified and hairpin-structured oligo DNA is worthy of development and assessment (48).

The flexible design (length, composition, etc.) of the oligo sequence (49) and diverse modification chemistry (fully methylated or hemimethylated to 6mA, 5mC, 4mC, etc.) of the oligo synthesis (50) would allow the MB probe to be used not only for the major Type II R-M system but also for many other types of R-M systems, such as Type I, Type III, and some minor subgroups of Type II (e.g., Type IIG) systems. Considering highly degenerate motifs often found in putative R-M systems, a mixture of oligos containing mixed bases would be preferred as a cost-effective choice. The so-called degeneracy of the probe in the context of the Type I R-M system refers to the unspecific region (typically $N_{2\sim10}$) of the bipartite sequence motif, while the degeneracy of the probe in the context of the Type III R-M system refers to the specific distance (e.g., $N_{25/27}$ for EcoP1I) between the recognition and cleavage sites. Except for such degenerate oligo

pool, two non-degenerate oligos, representing the highest and lowest frequency of a predicted sequence motif, would also be worthwhile to include in the enzyme assays. The design and use of those three kinds of MB probes can cover the full substrate spectrum of the enzyme activity toward any predicted degenerate motifs. It should be noted that the Type III R-M system is still active toward cleaving one-site target DNA (Fig. S9A), although its REase activity is lower than toward two-site ones (51). On the other hand, the MTase activity of the bifunctional Type III system is independent of the cleavage-associated degeneracy, meaning no need to include consecutive unspecific bases in the probe for the MTase assay, as exemplified with EcoP15I (Fig. S9B). The REase activity and MTase activity of other bifunctional R-M systems (e.g., Type IIG) can also be individually interrogated with the MB probe and MTase-Glo assay, respectively, via controlling the reaction solution composition (e.g., the presence or absence of SAM, ATP, specific ions, and so forth) (52).

In general speaking, short DNA oligos can avoid forming substantial unexpected secondary structures other than the desired hairpin structure. A probable issue associated with the oligo containing multiple degenerate bases is the formation of undesired secondary structures, albeit less stable than the primary hairpin structure. For example, the recognition sequence (CCC-N$_5$-RTTGY) of a 4mC/6mA Type I R-M system PacII (53) can generate $4^5 \times 2 \times 2 = 4,096$ combinations in an oligo pool, some of which might form metastable secondary structures. Optimizing the length and composition of the oligo sequence can reduce the probability of the formation of undesired secondary structures (e.g., the 78-nt-long MB probe in Fig. S9A possesses the sole hairpin structure as calculated by Oligo 7), but manual check of every individual oligo is impractical. We may think of a solution based on a software program capable of scoring the likelihood of undesired secondary structures and optimizing the oligo design (16).

Methylated, hydroxymethylated, or glucosyl-hydroxymethylated MB probes can be used for the modification-dependent Type IV system (54); however, the probe design might be rather complex than other types when the recognition sequence specificities are sometimes poorly defined.

## Variations and considerations in R-M studies based on the reconstituted cell-free system

The simple configuration of the PURE system has pros and cons. The lack of apparent nuclease and methyltransferase activities in the PUREfrex system allowed direct use of the CFPS reaction solution for the R-M enzymatic assays without complicated protein purification. Simply mixing different gene products in a well may also allow for protein-protein interaction studies, such as identifying valid combinations of specificity, restriction, or methylation subunits of Type I, III, or IV R-M enzymes, which are generally cumbersome in cell-based expression. However, it should always be taken into account that the PURE-based CFPS technology may not be able to synthesize some proteins of interest, probably due to the lack of co-translational or post-translational modifications or necessary cofactors. Therefore, a negative result can be attributed to either pseudogenization or a failure of protein synthesis. Collecting indirect evidence with bioinformatics tools is still an important approach complementary to direct biochemical characterization, and our present work emphasized a seamless connection between them.

A dialysis membrane pore blockage by protein adsorption or sedimentation can be a primary concern for the PURE system composed of high concentrations of protein components. Actually, a difference in the dialysis efficiency among several candidate microdialyzers has also been observed during the study. Given the diverse and proprietary configurations and materials, a preliminary screening for the candidate dialyzers is highly recommended in practice. As many microdialyzers are compatible with or directly manufactured in a microplate format, a parallel sample treatment can be readily envisioned with the aid of proper instrumentation (55, 56).

The sequence-dependent MTase assay based on G($^{6m}$A)TC-specific DpnI showed excellent compatibility with the PURE system. Other sequence specificities (e.g., the

physiologically important CG sites) may also be assayed so long as the corresponding methyl-directed REase is available (57). Like MB-based REase characterization, for MTase, we were further ambitious to expand the scope of the one-pot assay to arbitrary sequences and thus proposed combining the MTase-Glo assay with the PURE system. This combination would be extremely useful, particularly for the activity confirmation of putative MTases (e.g., three additional putative MTases of *Nitratiruptor* sp. YY08-14 provisionally registered as M.Nsp0814ORF1354P, M.Nsp0814ORF225P, and M.Nsp0814ORF70P in REBASE) without predictable motifs or methylation types. Additionally, the simple and well-defined composition of both PURE and MTase-Glo systems allows for straightforward modification of reaction components (e.g., metal ion cofactors) or reaction conditions (e.g., pH, temperature, and salt concentrations) to a large extent that may be required for elaborate kinetic measurement of new putative MTases. Kinetic parameters such as $V_{max}$ and $K_m$ can also be obtained with the MTase-Glo assay (38). Through systematically altering the reaction conditions mentioned above, a boundary between undesirable sequence promiscuity [known as star activity, an off-target effect dependent on reaction conditions (58, 59)] and well-defined degenerate (i.e., mixed bases: R, Y, S, W, K, M, B, D, H, V, and N) sequence specificity can be quantitatively defined.

## Difficulties and challenges in combination with fluorescent MTase assays

There have been several sequence-independent MTase assay methods based on the detection of methyl moiety or SAH with absorbance, fluorescence, or chemiluminescence in the literature (60). Due to potential applications in high-throughput screening of MTases with fluorescence microscopy (61), prior to establishing the combination of MTase-Glo and PURE system, we first tried a promising fluorescent MTase assay, SAM-fluoro, by which the SAH can be converted to hydrogen peroxide ($H_2O_2$) and detected with the Amplex Red chemistry (62, 63). Unexpectedly, the result showed high background fluorescence, resulting in two orders of magnitude decrease in the apparent signal-to-background ratio, from over 13 (Fig. 3E) to just 0.14 (Fig. S10). The probe specificity of Amplex Red and the artifactual generation of $H_2O_2$ in the presence of Amplex Red and peroxidase have long been questioned (64). Our measurement confirmed similar outcomes in the context of the cell-free system, although its composition is much simpler than cells, tissues, or clinical samples. Other fluorescent probes directly targeting $H_2O_2$ without coupling the peroxidase catalysis may be a promising alternative for MTase assays (65, 66). Another promising approach that we have considered compatible with the one-pot scheme is AlphaLISA or LANCE Ultra (67), a homogeneous and wash-free immunoassay directly targeting the methyl moiety. This kind of immunoassay has been used for some protein MTases but few for the DNA ones, probably due to the absence of universal antibodies or antibody mimics that can specifically bind to the methyl moiety on an arbitrary DNA strand. The efficient generation and selection of binding partners, including but not limited to antibodies, against the methylated nucleobases is still a great challenge for molecular engineering.

## $K_D$ issue toward single-molecule REase activity measurement

With respect to the Type II REase that generally forms dimers to function (23), an issue that should be carefully examined is the equilibrium dissociation constant ($K_D$) of the oligomeric active state. In a previous study (61), we proposed a limiting dilution of the CFPS solution of dimeric alkaline phosphatase, which enabled an encapsulation of single enzyme molecules into femtoliter compartments (68) and rapid measurement of the enzyme activity without protein purification. However, the same approach is not readily feasible for most REases due to the dissociation of the oligomeric active enzyme into inactive monomers, which is attributed to their high $K_D$ of generally nanomolar levels (69, 70). A single molecule in a femtoliter reactor can only result in picomolar concentrations. When nanomolar or sub-micromolar concentrations are required for stable oligomerization, the volume of the reactor must be reduced to attoliters ($10^{-18}$ L)

or sub-attoliters. It presents a technical challenge to microfabrication as the resolving power of traditional top-down lithography we applied before would become insufficient (71).

## Perspectives on investigation of R-M molecular diversities

Given the small repertoire of commercially available MTases, M.Nph3I surely expanded the arsenal of DNA methylation tools for potential biotechnology applications. The discovery of new enzymes with improved catalytic properties or novel substrate specificities not only provides practical research tools but also offers original research materials for elucidating the molecular principle behind them. For instance, M.Nph3I may act as a model MTase to study the ternary molecular interaction between MTase, DNA, and methyl donor that remains largely unknown. The by-product of methylation reaction, SAH, has a broad inhibition spectrum for MTases (72), which is known as product inhibition in enzymology and might also explain the incomplete DNA methylation by the *E. coli* Dam MTase. Occasionally, we found the rare exception that M.Nph3I was nearly completely liberated from the product inhibition toward the GGATCC motif. This enzyme is thus intriguing since just a single base change in the substrate led to the significant tolerance of high concentrations of SAH. The continued accumulation of relevant insights may enable a rapid and rational design of selective inhibitors or activators against a given MTase. Other examples such as R.Nph3I, M.Nph3II, and their respective isoschizomers or homologs are also undoubtedly good input materials to output the protein sequence-function relationship, an abiding interest in the field of protein science (73). Particularly for the R-M enzymes, the subtle and precise determination of extremely diverse DNA sequence specificities remains a mystery (74). In this sense, R.Nph3I and BstYI, which exhibited opposite methylation sensitivity but identical cleavage kinetics (Fig. 1B and E), can be an ideal pair to study the evolutionary relationship between methylation sensitivity and recognition specificity for REases since both enzymes are classified into the same superfamily in the InterPro database and probably close to each other in protein structure (25, 75). A continued exploration of real protein examples that can be accelerated by our one-pot approach may lead to an ultimate on-demand creation of R-M enzymes with arbitrary sequence specificities.

## Conclusion

In conclusion, we established an integrated solution based on the reconstituted cell-free system to confirm the activity and recognition sequence specificity of putative REases and MTases. The molecular beacon technology and the MTase-Glo assay are applicable to the one-pot REase assay and one-pot MTase assay, respectively. No protein purification is required for the activity confirmation. The sequential one-pot characterization workflow lightened the overall workload (pipetting is the only required operation) and shortened the probable trial-and-error period (from weeks or longer to days). As exemplified with the R-M enzymes from the deep sea in this study, there is a hidden world of many unexplored molecular diversities in diverse natural environments. Given the explosive increase of genetic information across all domains of life, there is a tremendous opportunity to acquire functional enzymes with superior properties distinct from the existing known ones, just as we occasionally found in this study. The simple experimental configuration compatible with microtiter plates and the small reagent consumption down to microliters lowered the technological and financial barriers to entry in high-throughput screening applications.

## MATERIALS AND METHODS

### Plasmid preparation

The DNA fragment of coding sequence of the putative REase NRS3_08 was PCR amplified (Platinum SuperFi PCR Master Mix, Invitrogen, Carlsbad, CA, USA) from the phage NrS-3

genomic DNA, which was purified in our previous study (21), and inserted into a T7 expression vector pET-3a (Merck, Darmstadt, Germany) via In-Fusion Cloning (In-Fusion HD Cloning Kit, Takara-Bio, Kusatsu, Japan) with two sets of primers (Table S2). *Escherichia coli* JM109 competent cells (RBC Bioscience, New Taipei City, Taiwan) were used for the plasmid transformation. The plasmid DNA was extracted (QIAGEN Plasmid Mini Kit, Qiagen, Hilden, Germany), and the sequence was checked by Sanger sequencing (FASMAC, Atsugi, Japan). The DNA concentration and purity were measured using NanoDrop One (Thermo Fisher Scientific, Waltham, MA, USA). The plasmid from the JM109 strain [$dam^+$, $dcm^+$, $hsdR17(r_K^-, m_K^+)$] is considered to be methylated.

## Preparation of linear DNA template

An overlap extension PCR (32) was designed and used to prepare the linear template DNA for a putative REase mel_015 (34) and two putative MTases NRS3_07 and C0176 (21), respectively (Fig. S3). A universal DNA fragment (namely, Fragment 1) containing 5′ UTR (untranslated region, including T7 promoter and ribosome binding site) was PCR amplified (30 cycles) from the T7 expression vector pET-3a with a proper primer set (P1F and P1R; Table S3). Another DNA fragment (Fragment 2) containing the CDS of target protein was also PCR amplified from the respective template DNA (synthetic gene for mel_015, viral DNA for NRS3_07, or bacterial genomic DNA for C0176) with another set of suitable primers (P2F and P2R; Table S3). The 5′ end of the P2R primer added 14 arbitrary nucleotides downstream of the stop codon. Fragment 1 and Fragment 2 were confirmed and recovered using E-Gel CloneWell II Gel (Invitrogen). Since the 5′ end of primers P1R and P2F was designed to overlap with each other, there was an overlapped region between Fragment 1 and Fragment 2. Equal moles of those two fragments in a total mass of about 10 ~ 20 ng were mixed and subjected to a short PCR (15 cycles) without primers. The PCR solution of 0.1 µL was directly added to a fresh 50 µL PCR solution and amplified with the forward primer P1F and reverse primer P2R. The PCR amplicon was confirmed by agarose gel electrophoresis. The PCR product was column purified (QIAquick PCR Purification Kit, Qiagen, Hilden, Germany).

## Cell-free protein synthesis

The *in vitro* protein expression was conducted in a standard microtube (FastGene 0.2 mL PCR tube, NIPPON Genetics, Tokyo, Japan). The template DNA of the putative REase (NRS3_08, mel_015) or the putative MTase (NRS3_07) in 3 ng/µL per 1 kb DNA was added to a reconstituted cell-free protein synthesis solution (PUREfrex 2.0, GeneFrontier, Kashiwa, Japan) containing 10 µL Solution I, 1 µL Solution II, 2 µL Solution III, and nuclease-free $H_2O$ (NIPPON GENE, Toyama, Japan) in a total volume of 20 µL. For another putative MTase C0176, since disulfide bond formation in this protein was strongly suggested by DiANNA (76), PUREfrex 2.1 was used in combination with glutathione disulfide (GSSG) and disulfide bond isomerase DsbC (GeneFrontier). The 20 µL reaction solution was composed of 8 µL Solution I (without cysteine and reducing agents), 1 µL cysteine, 1 µL glutathione (GSH), 0.33 µL GSSG, 0.25 µL DsbC, 1 µL Solution II, 2 µL Solution III, template DNA of 3 ng/µL per 1 kb, and nuclease-free $H_2O$. All the CFPS reactions were performed at 37°C for 4 h on a PCR thermocycler (Veriti, Applied Biosystems). Every resulting CFPS solution was stored at −80°C until use.

## Restriction endonuclease activity measurement

The REase activity was detected using molecular beacon technique (28). A set of MB probes (namely, MB-1 to 4; Fig. S2A) was designed and synthesized (FASMAC) for NRS3_08 (MB-1 probe: 5′FAM-cgAGATCTagttttctAGATCTcg-3′BHQ1; MB-2 probe: 5′FAM-cgGGATCCagttttctGGATCCcg-3′BHQ1; MB-3 probe: 5′FAM-cgAGATC-Cagttttctctctct GGATCCcg-3′BHQ1; MB-4 probe: 5′FAM-cgGGATCCTagttttctAGATCCcg-3′BHQ1). The predicted restriction site RGATCY was written in uppercase letters. The nucleobases (written in lowercase letters) flanking RGATCY were designed to be all the same across

these four probes to avoid unwanted potential effects of the flanking sequence on the digest efficiency (77). The formation of the stable ($\Delta G < -10$ kcal·mol$^{-1}$, $T_m$ >76°C) hairpin structure was confirmed by Oligo 7 (Molecular Biology Insights, Colorado Springs, CO, USA). The probe concentration was measured based on OD$_{260}$ and given by the oligo manufacturer upon delivery. The CFPS solution of 1.5 µL was mixed with 11.4 µL H$_2$O, 1.5 µL 10× NEBuffer 3.1 (New England Biolabs, Ipswich, MA, USA), and 0.6 µL of 500 µM MB probe (20 µM in final) in a 384-well µClear black non-binding microplate (Greiner Bio-One, Kremsmünster, Austria). The fluorescence signal was recorded using a plate reader (Tecan Infinite F200, Männedorf, Switzerland) at 37°C with a filter set of 485/20-nm excitation and 535/25-nm emission wavelengths.

## Preparation of methylated molecular beacon probes

N6-methyladenine (6mA)-modified (at the RGATCY motif) MB probe was chemically synthesized using 5′-dimethoxytrityl-N6-methyl-2′-deoxyadenosine, 3′-[(2-cyanoethyl)-(N,N-diisopropyl)]-phosphoramidite (Glen Research, Sterling, VA, USA) by FASMAC.

Alternatively, the MB probe in a final concentration of 50 µM was mixed with 1.12 mM S-adenosylmethionine (SAM) (New England Biolabs), 1× Dam methyltransferase reaction buffer (New England Biolabs), 16 U *E. coli* Dam methyltransferase (New England Biolabs) or 2 µL of the NRS3_07 CFPS solution, and nuclease-free H$_2$O to a total volume of 20 µL. The methylation reaction was carried out at 37°C for 6 h.

## Methylation sensitivity assay

The DNA methylation sensitivity of the REase was determined using the methylated MB probes. The continuous fluorogenic assay was carried out in a total volume of 15 µL at 37°C containing 1 µL of the REase CFPS solution, 10 µM methylated MB probe, and 1× NEBuffer 3.1. The fluorescence intensity was recorded using the plate reader (Tecan) with the same parameter settings used for the REase activity measurement. The DNA cleavage reaction rate was determined by linear regression of the initial linear region of the fluorescence intensity curve.

## Microdialysis

The CFPS solution of the target MTase (NRS3_07: 24.4 kDa; C0176: 41.9 kDa; the molecular weight was calculated by Expasy at https://web.expasy.org/compute_pi/) was subjected to dialysis against 0.1 M Tris-HCl (pH 8.0) buffer prior to MTase activity assay. In brief, the CFPS reaction solution of 10 ~ 20 µL was dialyzed using Tube-O-DIALYZER Micro 15K MWCO (G-Biosciences, MO, USA) on a mini stirrer (IS-M03, Ikeda Scientific, Tokyo, Japan) at 4°C. The CFPS solution was dispensed directly onto the dialysis membrane. It was crucial to keep the dialysis membrane wet until dispensing the sample on it. We dialyzed the sample overnight, although the dialysis time may be optimized according to the nature of the sample. The dialysis buffer was replaced every hour for the first 5 h and the last hour. The volume of the dialyzed sample increased two to three times due to an osmotic pressure-driven water inflow into the sample. The dialyzed CFPS solution was directly used for subsequent MTase activity assays without purification or concentration.

## Methyltransferase activity measurement

For putative MTase NRS3_07, a coupled enzymatic reaction was applied to detect the MTase activity (35). The reaction solution contained 1 µL of the MTase CFPS solution (not the dialyzed one), 10 µM MB probe, 320 µM SAM, 10 U DpnI (Takara-Bio), which only cleaves the methylated G($^{6m}$A)TC site, and the accompanying 1× T buffer (Takara-Bio) for DpnI. The assay in a total volume of 20 µL was carried out in a 384-well µClear black non-binding microplate (Greiner Bio-One) at 37°C using a plate reader (Powerscan HT, BioTek). The fluorescence signal was recorded with a filter set of 485/20 nm excitation and 528/20 nm emission wavelengths.

The MTase activity of NRS3_07 and C0176 was also detected with an MTase-Glo assay (Promega, Madison, WI, USA) (38). In brief, the dialyzed MTase sample of 1 µL was mixed with 10 µM DNA substrate (Fig. S2B and C) and 50 µM SAM in 1× reaction buffer (20 mM Tris-HCl, pH 8.0, 50 mM NaCl, 1 mM EDTA, 3 mM $MgCl_2$, 0.1 mg/mL BSA, 1 mM dithiothreitol), resulting in a total volume of 4 µL. The methylation reaction was carried out at either room temperature (23°C) or 37°C for 10 min to 3 h. Then, 5× MTase-Glo Reagent of 1 µL was added to the above reaction mixture, followed by additional 30 min incubation at room temperature, which converts the by-product of the methylation reaction, S-adenosylhomocysteine, to ADP. Lastly, the ADP was converted to ATP by adding MTase-Glo Detection Solution of 5 µL to the mixture, and the ATP was detected by luciferase in the mixture. The chemiluminescence reaction was carried out at room temperature in a 384-well low-volume white non-binding surface microplate (Corning, Tewksbury, MA, USA) for 30 min. An end-point luminescence measurement was carried out using a plate reader (Agilent BioTek Synergy H1M2, Winooski, VT, USA).

## ACKNOWLEDGMENTS

We thank Shigeru Shimamura (JAMSTEC) for the useful discussion regarding DNA methylation analysis and Biotechnological Research Support Division, FASMAC, and Life Science Solutions Group, Thermo Fisher Scientific, for the technical support regarding oligonucleotide concentration measurement.

This work was supported by a research grant from Kato Memorial Bioscience Foundation (to Y.Z.), JSPS KAKENHI Grant Number JP20K21455 (to Y.Z.), The Ministry of Education, Culture, Sports, Science and Technology (MEXT) of Japan (JP20K15444 to S.H.), ACT-X, Japan Science and Technology Agency (JPMJAX22BK to S.H.), and the budget of Japan Agency for Marine-Earth Science and Technology.

## AUTHOR AFFILIATIONS

[1]SUGAR Program, X-star, Japan Agency for Marine-Earth Science and Technology (JAMSTEC), Yokosuka, Japan
[2]Research Center for Bioscience and Nanoscience (CeBN), MRU, Japan Agency for Marine-Earth Science and Technology (JAMSTEC), Yokosuka, Japan

## AUTHOR ORCIDs

Yi Zhang  http://orcid.org/0000-0002-1886-737X

## FUNDING

| Funder | Grant(s) | Author(s) |
| --- | --- | --- |
| Kato Memorial Bioscience Foundation | | Yi Zhang |
| MEXT | Japan Society for the Promotion of Science (JSPS) | JP20K15444 | Satoshi Hiraoka |
| MEXT | JST | ACT-X | JPMJAX22BK | Satoshi Hiraoka |
| MEXT | Japan Agency for Marine-Earth Science and Technology (JAMSTEC) | | Ken Takai |
| MEXT | Japan Society for the Promotion of Science (JSPS) | JP20K21455 | Yi Zhang |

## AUTHOR CONTRIBUTIONS

Yi Zhang, Conceptualization, Data curation, Formal analysis, Funding acquisition, Investigation, Methodology, Project administration, Supervision, Validation, Visualization, Writing – original draft, Writing – review and editing | Yoshihiro Takaki, Conceptualization, Data curation, Investigation, Resources, Software, Visualization, Writing – review and editing | Yukari Yoshida-Takashima, Data curation, Formal analysis, Investigation, Resources, Validation, Writing – review and editing | Satoshi Hiraoka, Funding acquisition,

Software, Writing – review and editing, Data curation, Resources | Kanako Kurosawa, Data curation, Investigation | Takuro Nunoura, Resources, Supervision, Writing – review and editing | Ken Takai, Funding acquisition, Resources, Supervision, Writing – review and editing

## ADDITIONAL FILES

The following material is available online.

### Supplemental Material

**Supplemental material (mSystems00817-23-s0001.pdf).** Fig. S1 to S10; Tables S1 to S3.

### Open Peer Review

**PEER REVIEW HISTORY (review-history.pdf).** An accounting of the reviewer comments and feedback.

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
