## [Reviewer comments · mSystems]

A sequential one-pot approach for rapid and convenient characterization of putative restriction-modification systems

Yi Zhang, Yoshihiro Takaki, Yukari Yoshida-Takashima, Satoshi Hiraoka, Kanako Kurosawa, Takuro Nunoura, and Ken Takai

Corresponding Author(s): Yi Zhang, Kaiyo Kenkyu Kaihatsu Kiko

Review Timeline:

Submission Date:

August 3, 2023

Accepted:

September 5, 2023

Editor: Steven Hallam

Reviewer(s): Disclosure of reviewer identity is with reference to reviewer comments included in decision letter(s). The following individuals involved in review of your submission have agreed to reveal their identity: Pedro H. Oliveira (Reviewer #1); Richard J Roberts (Reviewer #2)

Transaction Report:

DOI: <https://doi.org/10.1128/msystems.00817-23>

September 5, 2023

Dr. Yi Zhang
Kaiyo Kenkyu Kaihatsu Kiko
Yokosuka, Kanagawa 2370067
Japan

Re: mSystems00817-23 (A sequential one-pot approach for rapid and convenient characterization of putative restriction-modification systems)

Dear Dr. Yi Zhang:

Thank you for your patience with the review process and your efforts to respond effectively to the suggestions and comments provided. Please ensure that when talking about Types of restriction enzyme, the word Type is capitalized (see Ref 31). This is a minor editorial comment but highly relevant to practitioners.

Your manuscript has been accepted, and I am forwarding it to the ASM Journals Department for publication. For your reference, ASM Journals' address is given below. Before it can be scheduled for publication, your manuscript will be checked by the mSystems production staff to make sure that all elements meet the technical requirements for publication. They will contact you if anything needs to be revised before copyediting and production can begin. Otherwise, you will be notified when your proofs are ready to be viewed.

If you would like to submit a potential Featured Image, please email a file and a short legend to mssystems@asmusa.org. Please note that we can only consider images that (i) the authors created or own and (ii) have not been previously published. By submitting, you agree that the image can be used under the same terms as the published article. File requirements: square dimensions (4" x 4"), 300 dpi resolution, RGB colorspace, TIF file format.

We recognize that the video files can become quite large, and so to avoid quality loss ASM suggests sending the video file via <https://www.wetransfer.com/>. When you have a final version of the video and the still ready to share, please send it to mSystems staff at mssystems@asmusa.org.

Sincerely,

Steven Hallam
Editor, mSystems

Journals Department
E-mail: mSystems@asmusa.org